# The Impact of Endometriosis on Pregnancy

**DOI:** 10.3390/jpm14010126

**Published:** 2024-01-22

**Authors:** Panagiotis Tsikouras, Efthimios Oikonomou, Anastasia Bothou, Penelopi Chaitidou, Dimitrios Kyriakou, Konstantinos Nikolettos, Sotirios Andreou, Foteini Gaitatzi, Theopi Nalbanti, Panagiotis Peitsidis, Spyridon Michalopoulos, Stefanos Zervoudis, George Iatrakis, Nikolaos Nikolettos

**Affiliations:** Department of Obstetrics and Gynecology, Democritus University of Thrace, 68100 Alexandroupolis, Greece; eftoikonomou@outlook.com (E.O.); natashabothou@windowslive.com (A.B.); penelopexai@gmail.com (P.C.); dimitriskyrkdk@gmail.com (D.K.); k.nikolettos@yahoo.gr (K.N.); soterisand@hotmail.com (S.A.); gkaitatzifoteini@hotmail.com (F.G.); theonalmpanti@hotmail.com (T.N.); peitsidiobgyn@gmail.com (P.P.); szervoud@otenet.gr (S.Z.); giatrakis@uniwa.gr (G.I.); nnikolet@med.duth.gr (N.N.)

**Keywords:** endometriosis, pregnancy complications, adverse pregnancy outcome

## Abstract

Despite the increased frequency of endometriosis, it remains one of the most enigmatic disorders regarding its effects on pregnancy. Endometriosis adversely affects both natural and assisted conception. Impaired folliculogenesis, which causes follicular dysfunction and low egg quality, as well as luteal phase problems, reduced fertilization, and abnormal embryogenesis, are some of the mechanisms advocated to explain reproductive dysfunction. There is a rising need for a comprehensive study of the potential negative consequences of this condition on pregnancy outcomes, including the postpartum period, as more women with a medical history of endometriosis become pregnant. Obstetrical complications (small for gestational age [SGA], cesarean section [CS], miscarriage, hemorrhage, low placental adhesion, and preterm delivery) are statistically elevated in women with endometriosis. Furthermore, ruptured ovarian endometrioma, appendicitis, intestinal perforation, and hemoperitoneum have been described in pregnancy. Obstetricians are largely unfamiliar with these complications, as they have not been thoroughly investigated. The development and pathogenesis of endometriosis is an important field of study and has not yet been fully elucidated. Finding these mechanisms is crucial for the development of new and more effective strategies to treat this condition. Endometriosis can have an impact on obstetric and neonatal outcomes of pregnancy, in addition to its potential effects on conception. To date, no additional monitoring is recommended for pregnancies with a history of endometriosis. However, more studies are urgently needed to assess the need for the tailored pregnancy monitoring of women with endometriosis.

## 1. Introduction

It is estimated that approximately 10% of women of reproductive age have some degree of endometriosis [1]. Although 75–90% of women experience reflux of menstrual material into the peritoneal cavity, only a small percentage of women of reproductive age develop foci of endometriosis [1,2,3]. Thirty percent of women with endometriosis suffer from infertility [1,2]. Other findings include delayed maturation of follicles and lower peak of endometrial thickness in the proliferative phase (“proliferative phase defect”) [4].

The most common localizations of pelvic endometriosis are the ovaries, the ligaments of the uterus, the Douglas pouch, and the fallopian tubes. Rare extrapelvic locations are the gastrointestinal system, presenting as hematochezia and periodic intestinal obstruction, the urinary system, with the most common symptoms being hematuria, dysuria, and pelvic fossa pain, the lung and pleura, manifesting as hemoptysis and chest pain, the peripheral nerves, appearing as dysesthesia and sciatica, and the diaphragm and the pericardium. Finally, cerebral endometriosis could result in perimenstrual headache and convulsions [5,6]. In certain cases, endometriosis can co-exist with endometrial and ovarian cancer [7]. Morphologically, three distinct types of endometriosis can be identified (red, white, and black), characterized by a progression of inflammation and fibrosis. Red lesions represent foci with increased vascularity and angiogenic activity. White lesions are an evolution of the red lesions that have undergone a process of inflammation and fibrosis. Classic black lesions are due to the healing process and formation of scar tissue [8].

Symptoms of endometriosis include heavy dysmenorrhoea, chronic pelvic pain, pain during and after intercourse, painful urination during menstruation, hypermenorrhea, slight leakage of blood between periods, infertility, and gastrointestinal problems, especially during periods, including diarrhea, constipation, and flatulence [7,8]. Endometriosis negatively impacts both natural and assisted conception [9,10]. Impaired folliculogenesis, which causes follicular dysfunction and low egg quality, as well as luteal phase problems, reduced fertilization, and abnormal embryogenesis, are some of the mechanisms advocated to explain reproductive dysfunction [11,12]. Endometriosis can have an impact on obstetric and neonatal outcomes of pregnancy in addition to its potential effects on conception [13,14]. The purpose of this paper is to review the literature on the relationship between endometriosis and pregnancy.

The literature search was mainly conducted by the authors PT, AB, and GI. A computerized literature search was executed to identify the pertinent studies reported in the English language. A comprehensive review of the literature from PubMed was undertaken from January 1980 to October 2023. This involved the keywords “endometriosis”, “adverse effects on pregnancy outcome” and “obstetric complications”.

## 2. The Pathogenesis and Pathophysiology of Endometriosis

Interacting endocrine, immune, proinflammatory, and proangiogenic mechanisms play a potential role in the development of endometriosis. The exact etiology and pathogenesis of endometriosis remain unclear.

Retrograde menstruation, metaplasia of the pit epithelium (the epithelium lining the abdominal organs), vascular and lymphatic metastatic spread, and neonatal uterine bleeding are the main hypotheses for the origin of endometrial cells in ectopic sites [15]. However, other elements, such as weakened or altered immunity, substances that promote angiogenesis, localized complex hormonal effects, and genetic factors, are required to increase cell survival, proliferation, and growth and maintain damage. According to the theory of metaplasia, the tissue of peritoneal mesothelial cells differentiates under the influence of hormones or inflammation factors [16,17,18,19]. In women without endometriosis, the regressed endometrium is destroyed by monocytes, and the remaining endometrial cells are destroyed by macrophages, natural killers (NKs), and cytotoxic lymphocytes. Endometrial cells are derived from reflux of menstrual material. Furthermore, under normal conditions, peripheral blood NK cells are able to autolyze endometrioid foci. In women with endometriosis, it was found that both local NK activity in the peritoneal fluid and blood NK cytotoxicity are reduced [20,21]. Of decisive importance is the finding that the more advanced the stage of endometriosis is, the greater the reduction in NK activity. An insufficient immune response of the peritoneum is reported in cases with endometriosis and in particular, an insufficient immune mechanism, local inflammatory response, increased concentration of interleukins, lysozymes, tumor necrosis factor (TNF), growth factors, and prostaglandins described in women with endometriosis intense angiogenic and neoangiogenic activity of the peritoneum, increased concentration and activity in the peritoneum angiogenic factors FGA (α-fibrinogen), TGF-a (transforming growth factor-a), TGF-b (transforming growth factor-β), HGF (hepatocyte growth factor), and VEGF-a (vascular endothelial growth factor-a) [22,23]. In the peritoneum of patients with endometriosis, increased concentrations of proinflammatory factors, such as interleukin 6, have been found [24]. In addition, the concentrations of factors favoring neoangiogenesis are also increased. These factors facilitate the attachment, survival, and growth of foci. Women with first-degree relatives with endometriosis have a seven-fold increased risk of developing the disease. At the same time, twin and family studies have identified the genetic burden. The genetic mechanisms involved in the genetic burden of endometriosis appear to be multifactorial or polygenic. Of course, researchers have not been able to isolate the genes involved and study their mode of action against female infertility. Understanding gene regulation and expression and how these processes depend on the presence of cells in ectopic sites is essential to understanding how the misfolded fragments cause the disease [25,26,27]. Specific interactions between the menstrual endometrial pieces and the peritoneal surface, however, are still somewhat controversial. One study suggested that endometrial stromal and epithelial cells can pass through the intact mesothelium, “Mesothelial inclusions in the ovarian cortex have the potential to transform into endometriosis through metaplasia”, while another one suggested that the adhesion of menstrual pieces only occurs when the underlying extracellular matrix of the mesothelium is exposed to local damage [11,28]. A large number of focused studies have examined the differences in gene expression and epigenetic modifications between eutopic and ectopic endometria, involving specific genes or their regulation by miRNAs. It should be noted that the eutopic endometrium is thought to be the origin of the majority of endometriotic lesions [29,30]. The genes purported to be aberrantly expressed in the ectopic endometrium are elucidated in Table 1 [31,32,33,34,35,36].

The idea that endometriosis develops from the metaplasia of the pit epithelium, which is dedifferentiated from the mesothelium, has been proposed. Recent research shows that this process involves reprogramming pluripotent mesenchymal stem cells, which can develop into endometrial epithelial and stromal cells in ectopic locations. These cells can be produced in the bone marrow or from a location within the endometrium itself [30,31,32].

Although metaplasia may explain deep endometriosis in the rectal septum, some investigators argue that it is unlikely to be the dominant mechanism underlying superficial peritoneal disease because the coexistence rate of the various endometriotic lesions (deep endometriosis, superficial lesions, and endometrioma) is higher than would be predicted if the lesions had a separate origin. This pathway is further supported by morphological changes from the endometriotic lesions in the ovarian surface epithelium. The rare occurrence of endometriosis in locations other than the pelvis, such as the abdominal lymph nodes, lungs, brain, extremities, and nasal cavity, as well as cases of Müllerian duct agenesis (a congenital malformation in which the Müllerian duct does not develop), is also believed to be rooted in metaplasia [33,34]. This theory is further backed by the highly rare observation of male patients suffering from endometriosis [24,29].

According to the hypothesis of metastasis, endometrial cells and tissue fragments spread from the uterine cavity via lymphatics and ectopic sites to colonize them. This theory explains the rare frequency of extrapelvic endometriosis in women, which is supported by data showing the emboli of endometrial cells in sentinel lymph nodes [29].

A relatively recent concept suggests that endometriosis may originate from stem or progenitor cells that could be found in the retrograde uterine bleeding of newborns. This happens when hormonal changes occur abruptly after birth due to the removal of the placenta. Supporting evidence of this theory includes the occurrence of neonatal uterine bleeding in 5% of newborns, a small number of girls developing premenstrual endometriosis, and a significant portion of adolescents experiencing severe endometriosis. Significant research has focused on epigenetic alterations in endometriotic lesions compared to normal endometrial tissue, as well as in the endometrial tissues of patients compared to those of healthy individuals. However, it is worth noting that only a limited number of these findings have been consistently replicated. Additionally, it should be considered that endometriotic damage is caused by the ectopic environment rather than the pathophysiology itself [29,30,31,32,33,34]. Most epigenetic studies predominantly focus on the elevation of DNA methylation levels [29,30,31,32,33,34]. One common result of this elevated methylation is the suppression of genes in the endometrium that are typically active during the secretory phase of the menstrual cycle. This can impact processes like cell proliferation and invasion. Notable examples of such duplicated epigenetic changes involve genes such as Hox-A10 (HOXA10), progesterone receptor B (PRB, encoded by PGR), and E-cadherin (also known as cadherin 1, CDH1) [30,33,34,35]. Differential methylation affecting HOX gene clusters, nuclear steroid receptor genes, and the expression of the GATA family of transcription factors was found in stromal cells from the endometrioma and healthy eutopic endometria, which appears to enhance progesterone resistance in endometriosis [11].

The investigation of epigenetic pathways involving histone modification in connection with endometriosis lacks comprehensive understanding, particularly in examining the decreased expression of miR-9, a microRNA that naturally hinders the anti-apoptotic BCL2 gene. The complexities of miRNA research, marked by discrepancies, are ascribed to problems in study design, cellular diversity, and variations associated with the menstrual cycle. Additionally, the essential influence of ovarian hormones in shaping contemporary medical approaches for managing endometriosis is emphasized [36,37,38,39,40,41,42].

The pathogenesis of endometriosis is inextricably linked to sex steroid hormones. For this reason, its occurrence is more frequent in women of reproductive age and less in menopausal women who do not take hormone therapy, similar to the normal endometrium; the ectopic is regulated by estrogen. More specifically, these are found in increased response, as a result of which the development of endometriosis is enhanced [37,38,39].

Moreover, specific environmental pollutants, such as dioxin, commonly found in various foods, can lead to progesterone resistance and the development of endometriosis. Simultaneously, many researchers argue that progesterone also contributes to the origin of endometriosis. While progesterone’s role is to control and suppress the mitotic activity of estrogens in the endometrium during its secretory phase, this regulatory function does not occur in abnormal growth, which is explained either as a malfunction of progesterone receptors or as their insufficient expression.

Estrogens play a crucial role as growth stimulants for endometrial cells [32,33,34,35,36,37]. Certain environmental factors, including pesticides and harmful substances, are thought to impact the production and breakdown of estradiol in women with endometriosis, leading to irregular cell growth. Studies have revealed elevated levels of steroidogenic factor 1 (SF1) in endometriotic stromal cells. SF1, a transcription factor, enhances aromatase gene expression and converts androstenedione into estrone and testosterone into estradiol, potentially contributing to the abnormal growth observed in endometriosis [36,37,38].

In contrast, 17-hydroxysteroid dehydrogenase 2 (encoded by HSD17B2), which typically converts estradiol to its less potent metabolite, estrone, is not expressed in ectopic endometrial implants or ectopic epithelia [35,36,37,38]. Therefore, estradiol accumulates in the area, resulting in the development of an estrogenic environment around the endometriotic lesions. In women with endometriosis, up to 30% experience progesterone resistance; this occurs due to dysregulated progesterone receptors or altered progesterone signaling pathways in both normal (eutopic) and abnormal (ectopic) endometrial tissues. This phenomenon involves the disruption of various genes activated by progesterone, including 2HSD17B2, progestogen-associated endometrial protein (PAEP), and TOB1 (a member of the antiproliferative (APRO) family of proteins that controls cell cycle progression in several cell types). Progesterone receptor subtype B (PRB) is relatively suppressed in this process. TOB1 acts as a cell cycle inhibitor, while PAEP, also known as glycodelin, serves as an immunomodulatory protein and a key marker of well-functioning endometrial tissue. Both proteins play roles in the anti-inflammatory and anti-proliferative effects of progesterone in the healthy endometrium [40,41,42,43,44,45]. The imbalanced effects of estradiol heighten tissue adhesion properties, increase the activity of matrix metalloproteinases, and trigger an angiogenic response among the factors influenced by disrupted steroid activity. These factors are essential for the development of abnormal growth outside the uterus. Furthermore, endometrial stromal and epithelial cells possess receptors for FSH and genetic variations in FSHB, the gene encoding the FSH subunit of the glycoprotein dimmer, which has been linked to endometriosis [46,47,48,49].

Concerning the receptivity of the endometrium, even in endometriosis patients with mild disease, the implantation rate is apparently lower in affected women both during natural cycles and during assisted reproductive treatments (ARTs). A reduced capacity for decidualization or the receptivity of the endometrium in these women may be the cause of defective implantation. Compared with the endometrium of healthy women, the eutopic endometrium of patients with endometriosis exhibits a number of molecular and functional abnormalities. It is still unclear whether the changes in the endometrial pattern are the cause of infertility and the presence of ectopic lesions or vice versa, as well as the mechanism and the specific signal that causes changes in the endometrial microenvironment of women with endometriosis [50].

Endometrial receptivity and decidualization are regulated by hormonally controlled molecular mechanisms. The main hormone responsible for the transient sensitive phenotype of the endometrium, necessary for embryo implantation, is progesterone (P4) [50,51].

The endometrial response to P4 is characterized by the estrogen-dependent suppression of epithelial cell proliferation, the maturation of glandular secretory systems, and the differentiation of stromal cells into specialized endometrial cells. The lack of receptivity of the endometrium of patients with endometriosis may be due in part to a functional dysregulation of steroid hormone transmission, which includes an increase in E2-mediated cell proliferation, inflammation, and progesterone resistance [50,51,52]. In the endometrium of women with endometriosis and mice with induced endometriosis, there are dysregulated levels of both total endometrial progesterone receptor (PR) expression and the ratio of its isoforms (PR-A/PR-B). In addition, estrogen receptor 1 (ESR-1) levels are higher in the midsecretory phase endometrium of women with endometriosis-related infertility than in control subjects, although progesterone receptor expression levels are lower in these subjects [50,51,52]. Integrins are endometrial cell adhesion molecules produced during the receptive window and are, therefore, essential for successful implantation. It is interesting to note that the endometrium of women with endometriosis has the abnormal expression of the B3-integrin subunit gene, which is a direct downstream target gene of both HOXA10 and ESR-1. Furthermore, in the clinical setting, ART is less successful in individuals who have lower levels of B3-integrin expression in the eutopic endometrium [51].

Other transcription factors (IGFBP1, GATA2, FOXO1, ARID1A, NOTCH1, and WNT4), involved in the control and mediation of progesterone signaling and essential for successful implantation, are similarly reduced in the endometrium of women with endometriosis [53]. 

## 3. The Impact of Endometriosis on Pregnancy

Several cases of acute pregnancy-related endometriosis problems have been reported. These problems, which can be life-threatening to both the mother and the fetus, are rarely reported and, therefore, underestimated [54,55,56,57,58]. 

This underscores the importance of healthcare professionals attending to pregnant patients with endometriosis and being mindful of these specific side effects. The following variables may be largely responsible for the complications of endometriosis during pregnancy [54,55]. Table 2 presents an overview of factors in endometriosis complications [56,57,58,59].

Progesterone removal causes bleeding, and the refolding of damaged arteries, and progesterone signaling must be maintained for decimation to occur. The necrosis of the foci of the damaged ectopic endometrium near dilated utero-ovarian or parametrial vessels could lead to the dysfunctional rupture of these vessels and the bleeding of unpredictable severity in endometriosis, which is characterized by progesterone resistance and the suboptimal expression of target genes [56,57]. A growing body of clinical data indicates that endometriosis can have a detrimental effect on how pregnancy normally develops. Epidemiological studies have yielded conflicting findings, and there are not enough carefully designed prospective trials to shed further light on the potential relationship between endometriosis and adverse pregnancy outcomes [58,59].

Taken together, changes in the local endometrial environment in patients with endometriosis may underlie the numerous processes proposed as the basis of the model, suggesting that late-stage disorders follow an adverse ripple effect in the early stages. The delayed implantation and improper placement of the embryo may be due to progesterone resistance and insufficient uterine contractility, inflammation, activation of free radical metabolism, and altered junctional zone (JZ) may promote shallow trophoblast invasion and preterm delivery [60,61,62,63,64].

Disorders in pregnant women that may be due to preimplantation abnormalities and that may continue throughout pregnancy include:Progesterone resistance and gene dysfunction: endometrial progesterone resistance leads to the dysfunction of genes crucial for embryo implantation [56];Inflammatory processes and uterine contractions: systemic maternal inflammation is a potential cause of preeclampsia. Uterine contractions, influenced by endometrial waves, show heightened activity in endometriosis patients [56];Metabolic free radical stimulation: the increased production of reactive oxygen species in endometrial cells contributes to maternal endothelial dysfunction [56];Thicker changes in the JZ: women with endometriosis exhibit thicker changes in the inner third of the myometrium, potentially impacting placental development [56].

## 4. Obstetrical Complications Associated with Endometriosis

In cases of endometriosis, the excessive intrauterine activation of free radicals, the resistance of the endometrium to the selective effects of progesterone, and the thicker endometrial junctional zone could lead to obstetric complications during pregnancy [55]. Similarly, the inflammatory processes in the endometrium, the pathological contractility of the uterus, the destruction of tissues and vessels due to chronic inflammation, and the creation of adhesions could lead to obstetric complications [56]. Furthermore, the ectopic endometrium observed in women with endometriosis is a toxic environment for placentation and results in its disruption, with the risk of the premature termination of the pregnancy, delayed intrauterine growth, and low birth weight for gestational age (Table 3) [56].

The following obstetric complications related to endometriosis can be considered:

i. First-trimester abortion (miscarriage);

ii. Ectopic pregnancy;

iii. Premature labor/premature rupture of membranes;

iv. Small birth weight for gestational age (SGA);

v. Intrauterine growth retardation (IUGR);

vi. Hypertensive disease of pregnancy and pre-eclampsia;

vii. Placenta previa;

viii. Obstetric bleeding;

ix. Bowel complications;

i.Miscarriage

Santulli and his team, in a 2016 study looking to elucidate the relationship between endometriosis and miscarriage, found that in a group of 478 women with endometriosis, 139 reported having a history of miscarriage. In contrast, the group of 964 women without endometriosis only had 187 reported miscarriages. Thus, they concluded that the miscarriage rate of women with endometriosis is higher than in women without [65,66,67]. Furthermore, Zullo’s 24 studies in 2017 showed that endometriosis could be characterized as a decisive predisposing factor for first-trimester miscarriage [62]. However, miscarriage risk was not increased in women with endometriosis who conceived via IVF compared to those who conceived naturally. It is worth noting that Yang and his colleagues, in 2019, conducted research among 1006 women with endometriosis having a history of laparoscopy or laparotomy and with 2012 women without the disease. The common element among all participants was that they had undergone a first cycle of IVF, achieving singleton pregnancies. They concluded that the risk of miscarriage was marginally the same. Thus, a general conclusion from the above study is that the type of conception, natural or artificial insemination, is not a predisposing factor for miscarriage [67].

ii.Ectopic pregnancy

Ectopic pregnancy is defined as the establishment of the embryo outside the normal location, i.e., inside the endometrium. The rate of spontaneous conception reaches 1–1.5%.

A study led by Farland mentioned that affected women are at a risk up to six times higher than women without endometriosis in 2019 [61].

iii.Premature labor

Premature birth is considered birth before 37 weeks. It is a major problem of obstetrics and gynecology since it is related to neonatal complications, such as intraabdominal bleeding, necrotizing enterocolitis, respiratory distress syndrome (RDS), and diabetes mellitus (DM). Predisposing factors for premature birth are the following:-Multiple pregnancies;-Polyhydramnios;-Sexually transmitted diseases (STDs);-Smoking.

Berlac et al. showed that women with endometriosis have twice the risk of preterm birth compared to those without the disease [68].

A higher risk of premature birth has been documented in the literature [69,70,71]. Preterm delivery affects 5% to 15% of pregnancies and is linked to neonatal morbidity and subsequent adult diseases [72]. The increased expression of inflammatory cytokines, such as interleukin 6, interleukin 1-β, and TNF-a, contributes to the onset of labor (premature or not). These local and systemic inflammatory cytokines are the same factors involved in the endometrium of women with endometriosis and preterm labor. Additionally, the levels of prostaglandin E2, cyclooxygenase 2, and various cytokines are increased in endometriosis foci compared to a normal endometrium. PGE2, COX-2, IL-6, and IL 1-b activation resulted in higher levels of progesterone and cytokines in the peritoneal fluid, greater myometrial activity, and twice the risk of preterm birth in endometriosis-affected women as compared to the control group [72,73,74]. In endometriosis, IVF does not raise the chance of a premature birth [75]. There is a heightened risk for preterm delivery and an increased risk of premature rupture of fetal membranes [76,77]. Possible causes for the above-mentioned pathologies are:¬Change in JZ;¬Local or systemic inflammation;¬Increased intrauterine pressure.
iv.Small for Gestational Age 

The elevated likelihood of small for gestational age (SGA) births is defined as a birth weight below the 10th percentile. Potential causes include placental hypoperfusion, reduced placental volume, progesterone resistance, and chronic inflammatory processes associated with endometriosis.

v.Fetal Growth Restriction

Fetal growth restriction (FGR), formerly known as intrauterine growth retardation or restriction (IUGR), is the condition in which the fetus lags behind in development and nutrition compared to what is expected for the gestational age. When the ultrasound scan shows FGR, it is necessary to accurately determine the gestational age and confirm the diagnosis by determining its severity and its cause. Normal fetal development is initially characterized by continuous cellular hyperplasia, followed by hyperplasia and hypertrophy, and, finally, only by hypotrophy [56].

Intrauterine growth restriction (IUGR) is associated with increased perinatal mortality and morbidity, emphasizing the significance of early diagnosis, especially in younger pregnancies without fetal pulmonary maturity, as these fetuses often fall into the broader category of small for gestational age (SGA) with varying birth weights. The definition of SGA relies on establishing lower ‘normal limits’, typically determined by percentiles, such as the 3rd, 5th, and 10th, but the choice of these limits influences the statistical characterization of SGA, posing a concern across obstetrics and gynecology and other specialties [55].

For the definition, two standard deviations (standard deviation, SD) below the average weight at the corresponding gestational age can be used [78,79]. Based on all of the above, there seems to be a link between endometriosis and the IUGR, a fact proven by Greek researchers Nirgianakis and his colleagues in 2018, reporting that women with a history of endometriosis, regardless of the degree of penetration, are at increased risk of IUGR compared to healthy women [80].

vi.Hypertensive disorders of pregnancy and preeclampsia

It is a heterogeneous, multisystemic disorder that usually manifests itself after the 20th week of pregnancy. The prevalence is 5–7/100 pregnancies. It has sometimes been called toxemia, toxicosis, hypertension with albuminuria, and pregnancy-induced hypertension (PIH). It is the second-leading cause of maternal mortality in pregnancy and is the most frequent cause of the induction of labor and iatrogenic prematurity. The following three discoveries characterize preeclampsia:-Blood pressure measuring 140/90 mmHg or above;-Albuminuria higher than 1000 mg/L or 300 mg in a 24-h urine sample;-Swelling of the limbs due to low serum albumin.

The only known cure for the condition is still the delivery of the fetus and the placenta’s departure. According to the findings of Zullo and colleagues’ 2017 study, the illness is not a prerequisite for the development of preeclampsia. Berlac discovered that patients with advanced-stage disease typically exhibit preeclamptic signs more frequently than women without the condition in the same year [62,68].

The risk of pre-eclampsia in women with endometriosis shows no statistically significant increase (OR 1.04; 95% CI, 0.83–1.29), affecting 5% of pregnancies. Abnormal placentation, characterized by increased vascular resistance, coagulation activation, and endothelial cell dysfunction, is theorized to involve uterine junction zone (JZ) changes. While JZ thickness is greater in endometriosis, correlating with pathological trophoblast infiltration and preeclampsia, women with endometriosis do not have an elevated risk of pre-eclampsia. JZ measurements—normally 5–8 mm, >12 mm in adenomyosis, and >8 mm in endometriosis—indicate no heightened risk. The correlation between hypertension during pregnancy and endometriosis does not show a discernible variation in the risk of gestational hypertension (OR 0.90; 95% CI, 0.59–1.37) [62].

vii.Placenta Previa

The occurrence of placenta praevia is heightened four-fold in women affected by endometriosis [67]. A possible cause is abnormal uterine contractility, impacting blastocyst implantation position coupled with heightened resistance to progesterone [56]. The presence of endometriosis increases the likelihood of placenta previa (OR 4.038; 95% CI, 2.291–7.116) [81].

viii.Obstetric Bleeding

Painless vaginal bleeding during the second and third trimesters is a clinical symptom of abnormal placenta implantation in the lower region of the uterus, which covers or partially encloses the cervix (less than 20 mm). The incidence is 0.3–0.5%.

Antepartum hemorrhage is up to 80% more prevalent in women with endometriosis, and its incidence is ten-fold higher in those suffering from deep endometriosis. Postpartum bleeding occurs 1.3 times more frequently in afflicted women [55,82]. 

A potential pathogenetic mechanism is abnormal blastocyst implantation due to differentiated JZ, dysperistalsis, a fixed uterus, local inflammation as well as possible coexisting adenomyosis, which also contributes to it [56]. 

Postpartum hemorrhage is conventionally defined as blood loss greater than 500 mL following the completion of the third stage of labor. Postpartum hemorrhage is defined by the American College of Obstetricians and Gynecologists as cumulative blood loss greater than 1000 mL combined with hypovolemia symptoms and signs. Roughly one-third of women undergoing cesarean sections experience blood loss over 1000 mL. These studies have shown that the estimated blood loss is often approximately half of the actual blood loss [56]. For this reason, estimated blood loss that exceeds the average loss should alert the obstetrician to possible excessive bleeding. Various studies have been carried out in order to clarify the connection between endometriosis and obstetric bleeding. Thus, in a study, it was reported that affected women with deep endometriosis have a risk of bleeding up to 10 times more compared to healthy women. A study by Berlac showed that women operated for endometriosis are at increased risk of bleeding both during pregnancy and after delivery [68].

Finally, in 2018, Nirgianakis and his staff found that women with a history of endometriosis, regardless of the degree of penetration, are at risk of bleeding [80]. The reason for this is that the junctional zone becomes abnormally differentiated, resulting in increased local inflammatory factors, the disruption of uterine contractility, and adhesions in the pelvic floor, which multiplies oxidative stress, a condition explained as an imbalance of oxygen free radicals and antioxidants, which appears to influence the pathophysiology of endometriosis.

Li et al. (2017) [83] demonstrated a statistically significant elevation in the risk of postpartum hemorrhage within the women affected by endometriosis (adjusted odds ratio: 2.265, 95% confidence interval: 1.062–4.872) when compared to the control group [84].

In a retrospective cohort study conducted by Shmueli et al. (2019) [85] encompassing 61,535 deliveries eligible for scrutiny, out of which 135 cases manifested endometriosis, it was observed that women afflicted with endometriosis exhibited an elevated likelihood of experiencing a postpartum hemorrhage (adjusted odds ratio [aOR] 3.7, 95% confidence interval [CI] 1.6–8.5), a postpartum hemoglobin level falling below 10 mg/dL (aOR 2.03, 95% CI 1.31–3.14), and necessitating packed cell transfusion (aOR 3.66, 95% CI 1.94–6.91) [86].

A few cases of the spontaneous rupture of uterine arteries have been described, one of which was accompanied by a hemothorax. Certain authors have hypothesized that the persistent inflammation associated with endometriosis might render utero-ovarian vessels more prone to fragility. Alternatively, the adhesions resulting from endometriosis could potentially create additional tension in these vessels, especially when the uterus undergoes enlargement during pregnancy [87].

This occurs as a generalized inflammatory reaction within the pelvis. To clarify, oxygen free radicals, originating from oxygen metabolism, have an inflammatory effect on cells, resulting in the regulation of cell proliferation and the toxicity of their role. In the case of endometriosis, macrophages, erythrocytes, and ectopic endometrial tissue, through retrograde menstrual flow, act inductively in oxidative stress.

ix.Bowel Complications

We ought to be mindful of significant intestinal issues, particularly bowel perforation, in the third trimester of pregnancy or IVF stimulation among women with deep endometriosis. This implies that, at least in certain cases, the endocrine conditions of pregnancy do not hinder the progression of endometriosis. The prevalence of this complication remains uncertain, likely due to potential underreporting [88].

## 5. The Fertility Implications of Endometriosis

In cases of minimal to mild endometriosis (stages I and II), normal conception without therapeutic means is greater than 2%, while in moderate and severe endometriosis (stages III and IV), the rate of normal conception without therapeutic means is much lower and immediate treatment is required [83,89].

### 5.1. Minimal to Mild Endometriosis 

Endometriotic foci produce proinflammatory cytokines (IL-1b, IL-8, IL-6, and TNFa) that cause a pathological environment for the follicles, affect the motility of the fallopian tubes, activate a generalized inflammatory reaction, and attract macrophages. At the same time, the endometriotic environment is toxic to the gametes and the macrophages of peritoneal fluid, which phagocytose the sperm with a direct impact on fertility. Furthermore, the hostile environment affects the reception of the endometrium and the implantation of the embryo [23,83,85,89,90,91].

### 5.2. Moderate to Severe Endometriosis

All of the above apply and in addition to the following.

Due to the pathological anatomy of the pelvis due to adhesions and endometriotic foci, the egg is not released properly during ovulation.

The egg is not transferred properly between the fallopian tubes.

Sperm is not transferred, which leads to the dysfunction of the peritoneum.

In women with endometriosis, increased peritoneal fluid is observed, which has a large number of macrophages, prostaglandins, and proteases. This fluid has an inhibitor that interferes with the normal interaction between eggs and fallopian tubes, resulting in difficulty in egg, sperm, and embryo implantation. Placental abruption is associated with the premature rupture of membranes and occurs more commonly in deep endometriosis.

There is an increased chance of cesarean section (CS) (OR 1.57; 95% CI, 1.39–1.78) [62].

Women with endometriosis face up to double the risk of giving birth via cesarean section [62].

According to the literature, increased complications during CS are reported [92,93,94].

Other obstetric complications that are statistically significantly increased due to spontaneous hemoperitoneum and endometriosis include rare but life-threatening complications occurring in the second half of pregnancy and immediately after delivery. The symptoms include acute or subacute abdominal pain, hypovolemic shock, and fetal distress.

The reasons for the above-mentioned symptoms are based on spontaneous rupture of vessels (natural hypertrophy + endometriotic foci + adhesions that increase the tension in the vessels) and bleeding from the endometriotic foci.

The diagnosis is difficult to determine preoperatively. Differential diagnoses include placental abruption, uterine rupture, spleen/liver rupture, and the HELP syndrome.

Focuses at 90% occur in the posterior uterine wall and intrapelvic area.

In the endometrial region, elevated levels of IgG and IgA antibodies, along with autoantibodies targeting endometrial antigens, are noted. This leads to the pathological reception of the endometrium and a decrease in implantation. In women with endometriotic foci, the syndrome of the luteinized unruptured follicle (LUF) can be observed. It is a syndrome of the follicle and not of the corpus luteum. To clarify, there is luteinization of the follicle and differentiation into a corpus luteum (with parallel secretion of progesterone), but without it breaking down and preventing the release of the egg [95].

In addition, luteal phase insufficiency may be observed in female patients. It is a pathological function of the corpus luteum, which does not produce a sufficient amount of progesterone, resulting in the short duration of the secretory phase of the cycle. The percentage is present in 3–4% of infertile women, while this increases in women over the age of 35 or in women with a history of regular miscarriages. The factors that point to the existence of this syndrome are the following:-A luteal phase lasting fewer than 11 days following the use of a thermometric chart with progesterone values in the middle of a normal range;-A luteal phase that normally lasts 14 ± 2 days, while progesterone values are low in the middle.-Progesterone values greater than 5 ng/mL and less than 10 ng/mL in the middle of the luteal phase, i.e., 6–8 days before menstruation [82].

In order to have a diagnosis, an endometrial biopsy must be performed 1–3 days before a period. Thus, when the growth of the endometrium falls short by more than two days, this syndrome is identified. Finally, in addition to the above luteinizing disorders, women with endometriosis may experience pathological follicular development and pathological fluctuations of LH. For women grappling with endometriosis, assisted reproductive methods frequently emerge as a primary recourse, particularly in the presence of concurrent conditions impeding natural conception. In women with a single factor of infertility, the surgical solution of the endometriotic foci is recommended, and then the application of some artificial reproduction methods [82].

Insemination and in vitro fertilization are considered suitable methods of assisted reproduction in women with endometriosis. Initially, intrauterine insemination (IUI) is considered appropriate in patients with stage I and II disease after controlled ovarian stimulation. These have more chances of pregnancy than women who do not follow any invasive method.

ASRM has specific protocols applied to patients with endometriosis, tailored accordingly to the stage of the disease:-In women younger than 35 years with stage I and II endometriosis, waiting (free sexual intercourse on fertile days) with or without induction of ovulation is recommended.-In women older than 35, the action should be more immediate. Insemination with ovulation induction is recommended, and in the case of failure, in vitro fertilization.-IVF is recommended for women with advanced-stage endometriosis (III/IV) who do not achieve normal conception after surgery or are old (over 35 years old) [96].

It is worth mentioning that no specific protocol for ovarian stimulation is recommended for women with endometriosis, but the selection of the appropriate protocol is conducted in the same way as for women with any fertility factor. Thus, the GnRH antagonist protocol has better results than the GnRH agonist protocol in patients with stage I and II endometriosis. In addition, the long protocol of GnRH analogs is recommended in patients with advanced-stage disease (III and IV) [69,70,71,87,97,98].

IVF has better results than intrauterine insemination in women with advanced endometriosis.

According to a cohort (series) study conducted in the unit of reproductive medicine at a university hospital in America in 2012, a sample of 2245 infertile women with various stages of endometriosis or tubal infertility factors was collected [99]. Its purpose was to prove which method of artificial reproduction is suitable for infertile women. The results of this research showed that women with endometriosis had the same chance of pregnancy or live birth as women with tubal obstruction. However, the American Society for Reproductive Medicine reports that patients with stage I and II endometriosis had a reduced fertility rate, while patients with stage III and IV disease required a higher dose of FSH, and fewer eggs were collected upon ovulation [99].

The conclusion of this research showed that women with various stages of endometriosis have the same success rates either with the IVF method or with intrauterine insemination. Of course, this goes against the recommendations of the European Society of Human Reproduction and Embryology.

A 2015 Oxford Academic survey found that women with a history of endometriosis had different outcomes from artificial insemination. This review sought to understand how endometrioma removal surgery affects the result of IVF or IUI.

They also sought to ascertain how endometriosis affects the outcomes of artificial insemination techniques and to determine the effect of different surgical techniques on the outcome of in vitro fertilization or intrauterine insemination [100].

The search included 33 studies for meta-analysis. Compared to women without endometriosis but with another infertility factor who underwent IVF/ICSI, women with endometriosis had similar outcomes. In addition, compared to women without surgery, women who underwent surgical endometriosis before IVF had better outcomes, and the number of eggs retrieved was similar [100].

In summary, women with endometrioma who underwent IVF/ICSI had similar reproductive outcomes compared to those without the disease, although their cycle cancellation rate was significantly higher [100].

Regarding the surgical treatment of endometriosis, this did not change the outcome of IVF/ICSI compared to those who did not undergo any surgical intervention. Finally, considering that reduced ovarian reserve can be attributed to the presence of endometriotic foci per se and the potential detrimental effect of surgery, individualizing care for patients prior to IVF/ICSI may help to optimize outcomes [100].

A 2020 study that was published in the National Library of Medicine found that it was feasible to ascertain how severe endometriosis affected IVF/IUI. Samples from patients who had laparoscopic cystectomies, as well as infertile women with ovarian endometriomas, were collected [101]. The results showed that significantly fewer eggs were collected in women with an advanced stage of the disease, while at the same time, they had a lower fertilization rate. In summary, according to the evidence, the advanced stage of endometriosis is an inhibiting factor for both the egg retrieval procedure, artificial insemination methods, the fertilization of the pelvis, adhesions, changes in the fallopian tubes, and a dysfunctional immune system [102]. At the same time, endometriosis can lead to the inhibition of ovulation, corpus luteum deficiency, unruptured luteinized follicles, and hyperprolactinemia. All these conditions can cause embryo implantation problems. As a result, there is an increased chance of miscarriage [103]. 

There is no linear relationship between endometriosis stage and infertility, however, it appears that the stage of the disease is associated with normal pregnancy rates.

It was observed that infertile couples with treated mild endometriosis have natural conception rates of 2% to 4.5% per 30 days vs. 15–20% for couples without the disease. However, in moderate and severe cases of endometriosis, the corresponding percentages are <2%, and for this, it is considered necessary that treatment occurs before trying to achieve pregnancy [104].

In the literature, referred IVF conception is associated with an increased incidence of perinatal complications, such as prematurity (double the risk), low birth weight (a 70% increase), small for gestational age (SGA) birth (a 40% increase), cesarean section (a 55% increase), intensive care (a 30% increase), and a 70% increase in perinatal mortality even in singleton IVF pregnancies The overall risk of neonatal complications was estimated to be twice as high in IVF pregnancies, and was not found to be increased for ICSI with embryo cryotransfer. It was also equally as frequent as that in naturally conceived pregnancies where the parents had previously sought medical help due to infertility. It is noteworthy that two studies reported an increased and equal risk of preterm or low birth weight newborns between infertile women who conceive naturally and those who resort to IVF. Although three meta-analyses showed an increased (30–40%) risk of congenital anomalies for both IVF and ICSI, a meta-analysis that considered infertility as a confounding factor did not confirm the findings. However, in another study, the increased risk of congenital anomalies in IVF neonates was reduced to non-significant levels when parental infertility was controlled for in the analysis [105,106,107,108]. There are concerns that the use of sperm from an infertile man and microinsemination may increase the risk of the conception of embryos with chromosomal abnormalities, but there are still few relevant studies [109]. Concerns have also arisen about the effect of IVF on ‘imprinted genes’ that are created during meiosis, usually due to changes in the methylation of a specific allele, rendering them silent.

However, due to the rarity of the corresponding syndromes, the absolute risk at the population level is expected to be negligible and undetectable. IVF in women with endometriosis does not elevate the risk of preterm birth. There are no differences in obstetric bleeding, cesarean section, and pre-eclampsia between women with endometriosis who conceived naturally and those who conceived via IVF. Placenta previa is more frequent in women with endometriosis who experienced pregnancy after IVF. More and more women are achieving pregnancy with the help of assisted reproductive techniques. Statistically, there are increased obstetric complications in women with endometriosis (SGA, CS, bleeding, premature delivery, and low placental adhesion) [69,70,71,97]. These complications are a topic that has not been sufficiently studied and remains unknown to the majority of obstetricians. Further research is essential to evaluate whether distinct pregnancy monitoring protocols are warranted for women with endometriosis. Endometriosis is a particularly important disease of the female reproductive system, which can affect other parts of the body in addition to the organs of the female reproductive system. Its development and pathogenesis are a significant field of study and have not yet been fully elucidated. Having endometriosis can affect different factors related to pregnancy. Women with endometriosis may experience reduced fertility rates, which may be due to different etiologies, such as changes in ovulation and endometrial receptivity or oocyte viability and interactions. Additionally, changes in pregnancy viability may be observed through a number of different mechanisms. Finally, endometriosis during pregnancy can lead to increased chances of miscarriage or placenta previa. The exact frequency and the mechanisms by which they occur have not been precisely elucidated [55].

Identifying these mechanisms is pivotal for advancing the development of novel and more efficacious strategies to address this condition. 

## 6. Conclusions

Despite the high frequency of endometriosis, it remains one of the most enigmatic disorders regarding its effect on pregnancy. Endometriosis adversely affects both natural and assisted conception. Impaired folliculogenesis, which causes follicular dysfunction and low egg quality, as well as luteal phase problems, reduced fertilization, and abnormal embryogenesis, are some of the mechanisms advocated to explain reproductive dysfunction. As more women with a history of endometriosis achieve pregnancy, there is a greater need for a systematic review of the potential adverse effects of the disease on pregnancy outcomes, including the postpartum period. Obstetrical complications are statistically increased in women with endometriosis (SGA, CS, miscarriage, bleeding, premature delivery, and low placental adhesion). These complications are a subject that has not been sufficiently studied and remains unknown to the majority of obstetricians. The development and pathogenesis of endometriosis is an important field of study and has not yet been fully elucidated. Its exact frequency and the mechanisms by which the reported complications in pregnancy occur have not been precisely elucidated.

Finding these mechanisms is crucial for the development of new and more effective strategies to treat this condition. Endometriosis can have an impact on obstetric and neonatal outcomes of pregnancy, in addition to its potential effects on conception. More studies are urgently needed to assess the need for the tailored pregnancy monitoring of women with endometriosis.

## Figures and Tables

**Table 1 jpm-14-00126-t001:** Aberrantly expressed genes in the ectopic endometrium.

Physiological Process	Genes	Encoding Protein
Adhesion	*ITGB2*	β2 integrin
Adhesion	*ITGB7*	β7 integrin
Proliferation	*PDGFRA*	Platelet-derived growth factor receptor-α
Proliferation	*PRKCB*	Protein kinase C-β1
Invasion	*RLN1, RLN2, RLN3*	Matrix metalloproteinases and relaxin
Immune recognition	*DEFB4*	Defensin-β4A
Inflammatory response	*IL1B*	IL-1β
Steroid biosynthesis	*Multiple genes*	Multiple proteins
Biosynthetic response	*Multiple genes*	Multiple proteins
Angiogenesis	*VEGF*	Vascular endothelial growth factor
Angiogenesis	*ANGPT1*	Angiopoietin 1
Angiogenesis	*ANGPT2*	Angiopoietin 2

**Table 2 jpm-14-00126-t002:** Overview of factors in endometriosis complications.

Pathophysiology	Details
Uterus expansion during pregnancy [56]	- Adhesions increase the tension on the nearby tissues - Adhesion development in the endometriosis: disease/treatment - Inflammatory cascade: fibrin deposition and coagulation - Inflammatory cell influx - Imbalance in fibrin deposition and degradation - Abdominal/pelvic surgeries, especially cesarean sections - Postoperative adhesions causing problems (bowel and bladder) - Pregnancy-related problems in people without endometriosis
Chronic inflammation in endometriosis [56]	- Fragility of tissues and arteries - Improper inflammatory response leading to tissue fibrosis - Chronic inflammation and fibrosis in endometriotic lesions - Hormone-saturated pregnancy environment enhancement - Endogenous inflammatory processes in ectopic lesions
Degraded endometrial tissue invasion [57,58,59]	- Increased back pressure and risk of tissue rupture - Well-vascularized tissue with increased permeability and edema - Vascular remodeling, angiogenesis, and luminal diameter - Folding of the damaged endometrium around enlarged vasculature - Alternative explanation for mechanical occlusion-induced disease - Arterial rupture

**Table 3 jpm-14-00126-t003:** Mechanism of action and the pathophysiology of endometriosis on pregnancy outcomes.

Mechanism of Action	Pathophysiology
Excessive endometrial activation of free radicals	Oxidative stress: the most important factor in adverse pregnancy outcomes
Resistance of the endometrium to the selective effects of progesterone	The FOAX gene and IGF-II are disrupted in women suffering from endometriosis
Thicker endometrial junctional zone (JZ)	During normal placentation, the spiral arteries become large vessels that connect the placenta and uterus at the junctional zone, while during pathological placentation, these arteries do not exist or are under-functioning, as a result of which the uterus and placenta do not connect properly

## Data Availability

Not applicable.

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
