# Peer review of "The Impact of Endometriosis on Pregnancy"

_jpm, 2024, doi:10.3390/jpm14010126_

Round 1

Reviewer 1 Report

Comments and Suggestions for Authors

The authors of this review performed great work regarding discussion of the most recent studies on the relationship between endometriosis and pregnancy outcomes. I was pleased to read this paper.

Despite that I have some questions and concerns.

First, the authors didn’t indicate how the literature search was conducted. What literature database and for what period was used?

Second, the authors describe in great detail the pathogenesis of endometriosis. And most of this information have been already discussed in others review articles. Besides, the section «Receptivity of the Endometrium» seems to be unduly after discussion of pathogenesis of endometriosis and can be shortened and combined with the previous one.

The section regarding the association between endometriosis and pregnancy is a lit bit confusing due to describing not the association, but more estimated impact this disease in pregnancy.

The section « Obstetrical Complications Associated with Endometriosis Contrary to this Fact» needs to be more structured. Some of the information can be described as a table or visualized. The same needs to be done with section «Discussion». Also, this section should be renamed according to the main topic of this one.

This review is overwritten and needs to be shortened and should be better structured. A lot of described facts may be presented as tables or schemes. Also, the references should be properly placed. Concatenating references at the end of a paragraph is not acceptable, as is their absence.

Author Response

First, we have added specific information about the literature search.

Second, we have shortened the and combined the section about  <<Receptivity of the Endometrium>>.

We have changed the title to <<The Impact of Endometriosis on Pregnancy>>.

We have done some major changes concerning this topic.

We have constrained the text within feasible boundaries and restructured its composition by incorporating tables. Bibliographic references have been reviewed and modified as required.

Reviewer 2 Report

Comments and Suggestions for Authors

The purpose of Tsikouras et al was to review the literature on the relationship between endometriosis and pregnancy.

This is very interesting paper; however that is very wide subject and I am not sure if all aspects are explained:

1.     There is lack of Abstract

2.     The methodology of the paper is unclear and the Authors should include precise information about inclusion/exclusion criteria as well as basic key words used in literature research.

3.     I am not sure about the construction of the manuscript: Introduction and Discussion?

4.     I don’t understand first sentence in the introduction, it looks like t is not finished.

5.     The part “Pathogenesis and Pathophysiology of Endometriosis” is a bit chaotic presented. It would be worth to add some explanation at the beginning, maybe table or figure to summarize all the information.

6.     Please remember to add explanation of all abbreviation using in the text.

7.     It would be worth to emphasize the difference between ectopic and eutopic endometrium in comparison to health endometrium.

8.     It would be worth to add more information about coagulation problems in endometriosis and problems during pregnancy.

9.     It should be clarified which problems could be involved in infertility, implantation and pregnancy or miscarriage or all.

10.  Please, check the citing criteria. Also, you can find more current information about endometriosis.

11.  I am not sure about that fragment – 524 – 539 – What do the Authors mean? There are more examples like that one.

12.  Generally, there are a lot of information, however, it needs some extensive editorial work to make it much more readable.

Author Response

  1. We have added the abstract.
  2. We have added specific information regarding the literature search.
  3. We have slightly modified the structure of the manuscript.
  4. We have deleted the first sentence in the introduction.
  5. We have changed the part “Pathogenesis and Pathophysiology of Endometriosis”.
  6. Each abbreviation has its own explanation.
  7. We have already done it.
  8. We have added more information about coagulation problems in endometriosis and during pregnancy.
  9. It has been clarified which problems could be involved in infertility, implantation and pregnancy or miscarriage.
  10. We have changed the literature adding some recent information.
  11. We have added more examples.
  12. We have changed the structure of the manuscript within feasible boundaries.

Reviewer 3 Report

Comments and Suggestions for Authors

Endometriosis is a complicated disease with many complications and implications in women's health, especially pregnancy. This is a good overview. It may support the need to screen (non-invasively) individuals for endometriosis. 

1. What is the main question addressed by the research? - Reviewing Endometriosis and its relationship with obstetrical complications 2. Do you consider the topic original or relevant in the field? Does it
address a specific gap in the field? - This is a very important and relevant topic in the field. The significant impacts it can have on women's health and obstetrical complications justifies the possible need for non-invasive tests to diagnose endometriosis. 
3. What does it add to the subject area compared with other published
material? - There is an extensive amount of pathophys that is very interesting. The purpose was likely to explain the possible connection btwn all of the complications that may occur due to endometriosis.  
4. What specific improvements should the authors consider regarding the
methodology? What further controls should be considered? - The above being said (in 3), it would benefit from clarification regarding the 'association' (vs possible impact) of endometriosis and pregnancy outcomes. The paper is very interesting and it was enjoyable to read.   Comments on the Quality of English Language

Appropriate

Author Response

  1. Reviewing endometriosis and its relationship with obstetrical complications.
  2. This is a very important and relevant topic in the field. The significant impact it can have on women’s health and obstetrical complications justifies the possible need for non-invasive tests to diagnose endometriosis.
  3. There is an extensive amount of pathophysiology that is very interesting. The purpose was likely to explain the possible connection between the complications that may occur due to endometriosis.
  4. We change the title to “The impact of endometriosis on Pregnancy”.

Round 2

Reviewer 1 Report

Comments and Suggestions for Authors

Thank you to the authors for the edits made. But unfortunately this article still can’t be published. Some of the parts of this article are still unclear. Besides, there are a lot of issues regarding references (location in the text, absence some references and abuse of quotation of review articles). Detailed comments are provided below.

Line 93: Abbreviation TNF was already used.

Line 91: need to add the decoding of abbreviation.

Line 149, 150-153, 165, 175: References should be properly located.

Line 176-191: This part can be shortened due to the contradiction and insufficient data regarding miRNA.

Line 204: The sentence “Estrogens play a crucial…” will be more logical as a start of the next paragraph.

Line 221: need to add the decoding ща abbreviation.

Line 234-235: repeated «endometriosis patients» and «women with endometriosis»

Line 286: reference?

Line 291–318: This text can be presented as a scheme or table.

Line 319–328: references should be properly located into the text.

Line 343–378: It is a too big piece to quote one reference, taking into account, that it’s a systematic review and you are quoting the results of articles, which have been quoted there.

Line 400: the name of the Table is required.

Line 439: The structure of this part is still unclear. Why the discussion of each “complication” have different type of marks? Why is needed to separate the paragraph “ 4.1 Adenomyosis” into the discussion of preterm delivery? This information not really significant for this part.

Line 513–557: references should be properly located in the text.

Line 533 and 541: What is correlation hypothesis?

Line 531: Is this sentence finished?

Line: 534–538: Separation of the text is unclear. Should be “;” in the end of each sentence?

Line 534 and 547: this abbreviation already has decoding.

Line 547–550: what is about?

Line 557: Reference?

Line 558: Is this sentence finished?

Line 566: Is this name of paragraph? The following text in brackets shouldn’t be here.

Line 575–587: The using a review for the reference of such a big part of the text is unappropriated.

Line 601–616: It should be used original references, not a systematic review.

Line 670: The part “Discussion” should be properly named. Some information needs to be removed and the methods of diagnosis and treatment should be discussed.

Author Response

  1. On page 3, at lines 94-95, we have made the necessary correction.
  2. On page 4, at line 98, we have deleted the tumor necrosis factor (TNF).
  3. On page 4, at lines 155-182, we have revised the references and appropriately positioned them.
  4. On pages 4 and 5, we have replaced the section referring to miRNA (lines 183-199) with a concise paragraph (lines 200-207).
  5. On page 5, at line 221, the sentence "Estrogens play a crucial..." has been relocated to the beginning of the subsequent paragraph.
  6. On page 5, at line 229, 238-239, we have added the decoding of the abbreviations.
  7. On page 6, at lines 253, we have substituted the <<women with endometriosis>> with <<affected women>>.
  8. On page 7, at line 304, we have added the appropriate references.
  9. On page 8, at lines 312-340, we have transformed the text into a comprehensive table with a title <<Table 2: Overview of Factors in Endometriosis Complications>> (page 7, line 310).
  10. On page 8, at lines 341-359, we cited the appropriate references
  11. On page 9, specifically in lines 373-414, we have withdrawn the text and replaced it with the content from pages 8 and 9, lines 360-372, including the relevant citations.
  12. On page 10, at line 437, we have written the name of the table << Table 3: Mechanism of action and the pathophysiology of the endometriosis on pregnancy outcome>>.
  13. On page 11, at line 473, we have changed the structure of this part. We have removed the part about adenomyosis (line 543) and we have placed the correct type of marks. Also, on page 12, we have changed the structure of the paragraph about small for gestational age. In addition to this, on pages 12 and 13, at lines 558-585, we have changed the structure of the text.
  14. On pages 13 and 14, at lines 607-648, significant changes have been made and the proper references have been inserted.
  15. On pages 13 and 14, at lines 622-649, the text has been removed and a new paragraph has been written on page 13, at lines 607-617, explaining the lack of correlation between preeclampsia and endometriosis.
  16. On page 14, at line 625, the sentence has been deleted.
  17. On page 14, at lines 636-649, the structure of the text has been modified.
  18. On page 14, at lines 607-617, the decoding has been appropriately provided.
  19. On pages 13 and 14, at lines 622-649, the text has been removed and a new paragraph has been written on page 13, at lines 607-617, explaining the lack of correlation between preeclampsia and endometriosis.
  20. On page 14, at line 637, the appropriate reference has been placed.
  21. On page 14, at lines 672-675, we have provided a new paragraph ιn place of lines 666-671.
  22. On page 12, at line 557, we have removed the brackets.
  23. On pages 12 and 13, at lines 578-585, we have cited a small paragraph in place of 565-578.
  24. On page 15, at lines 676-691, we have placed the original references
  25. On page 16, at line 538, we have changed the name <<Discussion>> to <<Fertility Implications of Endometriosis>>. Also, on page 17, at line 97 a new reference has been place. Similarly, this has been done on page 17, line 825, on page 18, lines 838, 844, 854, 859, 867, 871, 877, 880, and on page 19, lines 887 and 904.